

**Petrogenesis and tectonic setting of late Paleoproterozoic diorites in the**
**Trans-North China Orogen**
Zhiyi Wang [a, b], Jun He [a*], Wolfgang Siebel [b], Shuhao Tang [a], Yiru Ji [a], Jianfeng He [a], Fukun Chen [a]
a: State Key Laboratory of Lithospheric and Environmental Coevolution, School of Earth and
Space Sciences, University of Science and Technology of China, Hefei 230026, China
b: Institute of Earth and Environmental Sciences, Albert-Ludwig University Freiburg, Freiburg
79104, Germany
*Corresponding author: jhe1989@ustc.edu.cn (J. He)





**Abstract:** The Xiong'er volcanic rocks and mafic dike swarms mark a significant
magmatic event after the amalgamation of the North China Craton (NCC) in the
Paleoproterozoic, yet their tectonic origins remain controversial. Several
Paleoproterozoic diorite intrusions have received widespread attention recently. Their
genesis and geological significance are crucial for understanding the evolution of the
NCC. In this study, we report zircon U-Pb ages and geochemical data of the
Jiguanshan diorite. The diorites in the Trans-North China Orogen, including the
Jiguanshan diorite, have comparable element and isotopic geochemical characteristics.
The weighted mean average of initial $^{87}Sr/^{86}Sr$ and $\varepsilon_{Nd}(t)$ values is 0.7052 ±0.0003
and -6.5 ±0.2, respectively. The initial Pb isotope compositions of these diorite
samples do not show significant enrichment of radiogenic lead. In terms of Sr-Nd-Pb
isotope compositions and Nb/Ta, Ba/Th, and Sr/Th ratios, these diorites differ from
the Xiong'er volcanic rocks and mafic dike swarms. Our results suggest that these
diorites originated from the basaltic lower crust, rather than from the enriched
subcontinental lithospheric mantle. Whole-rock and zircon trace element geological
tectonic diagrams indicate that the diorites formed in a rift environment. These
diorites mark a crustal-origin rock shift from orogenic-related magmatism to
intraplate magmatism during the post-collisional extensional stage.
**Key words:** Late Paleoproterozoic, North China, Diorite, Zircon, Sr-Nd-Pb isotopes

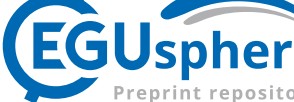

## 1 Introduction

The ancient basement rocks in the North China Craton (NCC) provide crucial insights into the Precambrian geological evolution (e.g., Geng et al., 2012; Liu et al., 1992). The main assembly of the NCC took place after the collision of eastern and western land masses in the late Paleoproterozoic (e.g., Zhao and Zhai, 2013; Zhao et al., 2000a, b). Subsequently, the craton experienced multiple rift phases, with the Xiong'er rift being the first rift formed after the assembly, resulting in the formation of the *c.* 1780 Ma Xiong'er volcanic rocks and contemporaneous mafic dyke swarms (e.g., Hou et al., 2008; Peng et al., 2007, 2008; Zhai, 2010). However, the origin and tectonic setting of the Xiong'er volcanic rocks and contemporaneous mafic dyke swarms of the NCC remains controversial. The debate mainly revolves around subduction (e.g., He et al., 2009; Wang et al., 2004; Zhao et al., 2009), rifting (e.g., Cui et al., 2010; Zhao et al., 2007), and the involvement of mantle plumes (e.g., Hou et al., 2008; Peng et al., 2007, 2008). Clarifying the tectonic setting during this period is essential for understanding the post-collisional orogenic evolution that followed the late Paleoproterozoic amalgamation of the North China Craton.

In recent years, numerous *c.* 1780 Ma diorites along the southern margin of the NCC and the Shanxi region (Fig. 1a) have attracted significant attention, potentially offering new perspectives for understanding the tectonic evolution of the craton during the late Paleoproterozoic. These rocks include the diorites intruding into the Xushan Formation (ca. 1789 Ma; Zhao et al., 2004), the East-West Group dykes (ca. 1780 Ma; Peng et al., 2007), the Shizhaigou diorite (ca. 1780 Ma; Cui et al., 2011), the Wafang diorite (ca. 1750 Ma; Wang et al., 2016), the Gushicun diorite (ca. 1780 Ma; Ma et al., 2023a), the Muzhijie diorite (ca. 1780 Ma; Ma et al., 2023b), the Fudian diorite (ca. 1780 Ma; Ma et al., 2023b), and the Jiguanshan diorite (ca. 1780



Ma; this study). These diorites are widely distributed, with similar zircon ages and an
approximate east-west trend. Some studies suggest some of them share a source with
the Xiong'er Group volcanic rocks or dyke swarms, formed by fractional
crystallization of enriched mantle (Cui et al., 2011; Peng et al., 2007). Others propose
some of them resulted from the fractional crystallization (Ma et al., 2023a, b) or
crustal melting with limited mantle influence (Wang et al., 2016). Systematic research
into their genesis is crucial for clarifying their formation and constraining regional
geological evolution.
The present study focuses on the Jiguanshan diorite and other diorites with ages of *c.*
1.78–1.75 Ga from the NCC. These diorites have similar geochemical characteristics,
suggesting they may have formed during a single magmatic episode. By evaluating
the geochemical and Sr-Nd-Pb isotopic compositions of whole rocks, as well as Hf
isotopic compositions of zircons, a better understanding of the tectonic environment
and evolution of the NCC during the late Paleoproterozoic is provided.

**2 Geological background and samples**
The NCC records a 3.8 Ga lasting geological evolution (e.g., Geng et al., 2012; Liu et
al., 1992). It was ultimately formed by the assembly of the eastern and western blocks
along the central orogenic belt at the end of the Paleoproterozoic (e.g., Zhao and Zhai,
2013; Zhao et al., 2000a, b). The southern margin of the NCC is mainly occupied by
the "Xiong'er rift", which is separated from the North Qinling Orogen by the
Luonan–Luanchuan fault (Fig. 1b). Before the Mesozoic, the southern margin of the
NCC experienced a similar geological evolution as the NCC itself, which makes it an
ideal object for studying the Precambrian geological evolution (e.g., Zhai, 2010).
The study area is located in the eastern part of the southern margin of the NCC (Fig.



1b). The most frequent basement rocks in this area are metamorphic basement rocks
of the Archean Taihua Group. The Taihua Group extends in an east-west direction
from Lantian in the west to Wuyang in the east (e.g., Diwu et al., 2014, 2018; Wang et
al., 2020). The upper part of the basement contains volcanic rocks of the Xiong'er
Group that formed *c.* 1780 Ma (e.g., Zhao et al., 2004, 2007). The Xiong'er volcanic
rocks consist mainly of basalts and andesites that are widely distributed along the
southern margin of the NCC, and extend as far north as Taiyuan City in Shanxi
Province (Zhao et al., 2007). The Xiong'er Group represents the largest magmatic unit
of the NCC since the Neoarchean period. At the same time, a large mafic dyke swarm
emplaced the NCC. These mafic rocks are interpreted as products of crustal extension
during the Colombia supercontinent era (e.g., Peng et al., 2008; Wang et al., 2004).
During our fieldwork, seven diorite samples were collected from the Jiguanshan
diorite on the eastern side of the Jiguanshan Hill (or the Jiguan Mountain), about 30
km south of Ruyang County, Henan Province (Fig. 1c and Table S1). The Jiguanshan
diorite forms several east-west striking bodies that are cut by the Mesozoic
Taishanmiao A-type granite to the west. The Taishanmiao intrusion, located at the
southern margin of the NCC in the western Henan region, covers an area of *c.* 290
km$^2$ (e.g., He et al., 2021). The northern and eastern part of the Taishanmiao intrusion
penetrates the volcanic rocks of the Xiong'er Group (Fig. 1c).
The collected samples of Jiguanshan diorite are fresh and greyish with massive
structures. They are fine-grained with a particle size of 0.1–2 mm (Fig. 2a, b). The
main mineral is plagioclase (~60 vol.%), which varies greatly in size and has a
lamellar and euhedral shape. Under the microscope, the partially sericitized crystals
show simple contact twinning and polysynthetic twinning. Some plagioclase crystals
show zonal and resorption textures (Fig. 2c-e) and the Carlsbad-Albite composite twin



with zoned texture (Fig. 2d). Clinopyroxene (~15 vol.%) formed earlier than
plagioclase. Most of the clinopyroxenes have zonal and resorption textures (Fig. 2f).
Euhedral opaque minerals (~3 vol.%), such as ilmenite, are often encased in
clinopyroxene. Alkaline feldspar (~10 vol.%) shows hypidiomorphic to xenomorphic
texture with imprints of kaolinization (Fig. 2c, e). The mineral occurs as potassium
feldspar and perthite. Quartz (~5 vol.%) can also appear as an anhedral crystal. Biotite
(~3 vol.%) shows xenomorphic texture or is altered into chloride (Fig. 2c, e). In
addition, accessory minerals such as zircon and ilmenite account for about 3 vol.%
(Fig. 2f).


**3 Method**
**Major and trace elements:** Seven representative fresh rock samples were selected to
be broken up into powders less than 200 mesh. Major element composition of whole
rock was obtained by X-ray fluorescence (XRF) from ALS Chemex (Guangzhou)
using a PANalytical PW2424 instrument. Following the sample digestion, whole-rock
trace element concentrations were determined using an Agilent 7700 inductively
coupled plasma mass spectrometry (ICP-MS) at the University of Science and
Technology of China (USTC). Quality control assurance was achieved by using GSR–
1, BCR–2, and AGV–2. The analytical uncertainties are <5%.
**Whole-rock Sr-Nd-Pb isotopes:** Chemical separation of whole-rock Sr-Nd-Pb
isotope analysis was performed in the ultra-clean laboratory of the Laboratory of
Radiogenic Isotope Geochemistry, USTC. Whole-rock powders of *c*. 100 mg were
weighed in 7 ml Teflon cups in a solution of purified HF and $HNO_3$ acids for Pb





isotopic analysis and in a solution of purified HF and HClO$_4$ acids for Sr-Nd isotopic
analysis. Sr and Nd were separated by AG 50W-X12 resin in 200–400 mesh purposes
and purified using the Sr-Spec® ion-exchange resin for Sr and Ln-Spec® resin for Nd.
All isotopic measurements were done on a Triton Plus mass spectrometer of Thermo
Scientific$^{TM}$. Measured Sr and Nd ratios were normalized to $^{86}$Sr/$^{88}$Sr = 0.1194 and
$^{143}$Nd/$^{144}$Nd = 0.7219, respectively. Pb isotope ratios were corrected for mass
fractionation using a fractionation factor of 0.1% per atomic mass unit based on
repeated measurements of reference material NIST NBS 981 (Wang et al., 2023a).
Total procedure blanks for Sr, Nd, and Pb were <200 pg. Detailed procedures can be
found elsewhere (Chen et al., 2000, 2007). The errors of the initial values of Sr and
Nd isotopes were obtained by the error transfer formula, which is shown in Table 2 for
Sr and Table 3 for Nd. Detailed formulas can be found in Siebel et al. (2005). A 5%
age error, a 2‰ $^{87}$Rb/$^{86}$Sr measurement error, and a 0.3‰ $^{87}$Sr/$^{86}$Sr measurement error
were used for the error of initial Sr values for calculation. A 5% age error, a 0.3‰
$^{147}$Sm/$^{143}$Nd error, and the $^{143}$Nd/$^{144}$Nd measurement error were used for the
calculation of the error of initial Nd isotope values.
**Zircon U-Pb age and trace elements:** Zircon crystals were isolated from the rocks
by standard mineral separation procedures. Grains with intact crystal shape and no
obvious inclusions were selected under a binocular microscope. The zircons were
embedded in epoxy resin. The upper and lower planes of each zircon target were
polished with sandpaper from coarse to fine. Most of the zircon gains were polished to
2/3 of the position and then cleaned in ultra-pure water by ultrasonic waves. The
grains were cleaned with dust-free paper in a certain direction to ensure that the zircon
was clean and bright without impurities under the microscope for carbon plating.
Cathodoluminescence (CL) image analysis was done on a scanning electron



microscope (SEM) located at the USTC. Zircon U-Pb isotopic and trace element
compositions were obtained by laser-ablation inductively-coupled plasma mass
spectrometry (LA-ICP-MS) using an Agilent 7700 ICP-MS with a 193 nm ArF
laser-ablation system at the USTC. The beam spot diameter was 32 μm, operating at a
repetition rate of 10 Hz. Helium served as the carrier gas. Zircon 91500 was used as a
standard for age calculation. The NIST SRM 610 and 612 were utilized as reference
materials for content adjustment. U-Pb ratios and uranium and lead concentration data
were calculated by the ICPMSDataCal software (Liu et al., 2010). Concordia and
weighted mean age plots were made using IsoplotR (Vermeesch, 2018).

**4 Analytical results**
Whole-rock compositions of the Jiguanshan diorite are given in Table 1, and
Sr-Nd-Pb isotope compositions and error calculations are shown in Tables 2 to 4. Age
results of zircon grains from four samples are given in Table S1, and trace elemental
contents in Table S2.

**4.1 Zircon U–Pb isotopic ages**
Zircon grains from the Jiguanshan diorite are transparent to pale yellow with
subhedral to euhedral habitus. They measure *c.* 100–300 μm in length and have aspect
ratios of 1:1 to 3:1. Most of them show oscillatory zoning in the CL images (Fig. 3),
which suggests their magmatic origin.
Twenty-nine zircon grains from sample ZY2202 yield $^{207}Pb/^{206}Pb$ ages varying from
1885 ±44 Ma to 1643 ±42 Ma and giving a weighted mean age of 1772 ±16 Ma (2σ,
n=29, Fig. 4a). Thirty-two zircon grains from sample ZY2204 yield $^{207}Pb/^{206}Pb$ ages



varying from 1902 ±54 Ma to 1635 ±47 Ma with a weighted mean age of 1742 ±15
Ma (2σ, n=32, Fig. 4b). Twenty-six out of twenty-seven zircon grains from sample
ZY2205 yield $^{207}Pb/^{206}Pb$ ages varying from 1933 ±52 Ma to 1692 ±44 Ma and a
weighted mean age of 1760 ±18 Ma (2σ, n=26, Fig. 4c). One zircon has a $^{207}Pb/^{206}Pb$
age of 1639 ±46 Ma (96% concordance), which is excluded from the calculation (Fig.
4c). Thirty zircon grains of sample ZY2207 yield $^{207}Pb/^{206}Pb$ ages ranging from 1900
±54 Ma to 1700 ±36 Ma with a weighted mean age of 1771 ±17 Ma (2σ, n=30, Fig.
4d).
Most zircon grains have Th/U ratios >1, supporting their magmatic origin (Table S1).
Some grains deviate from the concordant line, which is related to lead loss (Fig. 4a-d).
The weighted mean ages of the Jiguanshan diorite near 1780 Ma suggest that the
diorite body formed in the late Paleoproterozoic.

**4.2 Whole-rock chemical composition**
The $SiO_2$ contents of the Jiguanshan diorite vary between 55.57 and 59.44 wt. % and
the sum of $K_2O+Na_2O$ from 5.57 to 6.03 wt. %, corresponding to gabbroic diorite to
diorite composition according to the TAS diagram (Fig. 5a). $K_2O$ contents range from
2.97 to 3.21 wt. % and fall within the high-K calc-alkaline fields (Fig. 5b). The
samples from the Jiguanshan diorite have consistent A/CNK ratios ranging from 0.78
to 0.81 and A/NK >1, which classify them as metaluminous rocks (Fig. 5c). $Mg^{\#}$
($Mg^{\#}=(MgO+FeO_{total})/MgO\times100$) values range from 34 to 39 (Fig. 5d).
The Jiguanshan diorite depicts the enrichment of large ion lithophile elements (LILE),
such as Rb, Ba, and K, and negative anomalies of Sr, Ti, Nb, and Ta (Fig. 6a). ∑REE
contents range from 361 ppm to 393 ppm. Light rare earth elements (LREE) exhibit



stronger enrichment, while heavy rare earth elements (HREE) are relatively depleted
(Fig. 6b). Their $(La/Yb)_N$ ratios range from 12.2 to 15.0 (subscript N denotes
normalization against chondrite La and Yb contents) with $Eu/Eu^*$
($Eu/Eu^*=2Eu_N/(Sm_N+Gd_N)$, subscript N denotes normalization against chondrite Sm
and Gd contents) ratios ranging from 0.57 to 0.68 (Table 1).

**4.3 Whole-rock Sr-Nd-Pb isotopic compositions**

All initial radiogenic isotopic values and the errors of the initial values of Sr, Nd and
Pb isotopes are calculated back to an age of 1780 Ma. The measured $^{87}Sr/^{86}Sr$ isotope
compositions of Jiguanshan diorite samples vary from 0.715177 ±0.000011 to
0.724714 ±0.000012 (2σ). Initial Sr ratios range from 0.7020 ±0.0007 to 0.7058
±0.0010 (2σ, Fig. 7a). Measured $^{143}Nd/^{144}Nd$ values vary from 0.511129 ±0.000008 to
0.511329 ±0.000007 (2σ). Initial $^{143}Nd/^{144}Nd$ isotope compositions range from
0.509924 ±0.000061 to 0.510090 ±0.000063 (2σ), corresponding to initial $\varepsilon_{Nd}$ values
of -8.04 ±1.20 to -4.80 ±1.23 (2σ, Fig. 7b) and two-stage Nd model ages ($T_{DM2}$) of
2.94 Ga to 2.68 Ga. Their Pb isotopic compositions are as follows: $^{206}Pb/^{204}Pb =$
15.832–16.167, $^{207}Pb/^{204}Pb = 15.170$–15.243, and $^{208}Pb/^{204}Pb = 36.046$–37.324. Initial
Pb isotope ratios are significantly lower: $^{206}Pb/^{204}Pb_i$ ratios ranging from 14.965 to
15.295, $^{207}Pb/^{204}Pb_i$ ratios ranging from 15.090 to 15.150, $^{208}Pb/^{204}Pb_i$ ratios ranging
from 34.398 to 35.825, with $^{238}U/^{204}Pb$ and $^{232}Th/^{238}U$ ratios ranging from 2.3 to 2.9
and 5.3 to 7.8, respectively (Fig. 8a, b).

**5 Discussion**

**5.1 Compositional characteristics of late-Paleoproterozoic diorites of the NCC**



The late Paleoproterozoic diorites in the NCC have uniform east-west (EW) strike,
suggesting a possible correlation, which differs from the north-northwest (NNW)
strike of most contemporaneous mafic dykes (Hou et al., 2008; Peng et al., 2007,
2008). The intrusion ages of these diorites are concentrated between 1780 and 1750
Ma. All the diorites have similar geochemical and isotopic compositions and can be
regarded as a compositional homogeneous rock group.
Summarizing the late-Paleoproterozoic diorites of the NCC, most of them have $SiO_2$
contents in the range of 52–62 wt. % (Fig. 5a). Total alkali content ($K_2O+Na_2O$) of 5–
7 wt. % suggests a subalkaline character for the diorite rocks (Fig. 5a). The $K_2O$
contents of these samples range from 2–5 wt. % in accordance with a high-K
calc-alkaline to shoshonite series (Fig. 5b). The ASI and $Mg^{\#}$ values of these samples,
except for a few data points that deviate significantly, are mostly homogeneous, with
weighted average values of 0.81 and 37, respectively (Fig. 5c, d). On primitive mantle
normalization diagrams, all the diorites display enrichment of large ion lithophilic
elements (LILEs), such as Rb, Ba, and K, and depletion of high field strength
elements (HFSEs), such as Na, Ta, Th, U, and Ti (Fig. 6). On the rare earth element
normalization diagrams, they have negative Eu anomalies with enrichment in LREEs
and flat distribution of HREEs (Fig. 6). As can be seen from the above, the oxides and
trace elements of these diorites have similarities.
All diorites have similar Nd isotopic compositions with the mean initial $\varepsilon_{Nd}$ value of
-6.51 ±0.2 (2σ, n=41, Fig. 7b), when we recalculate the initial $\varepsilon_{Nd}$ values and their
errors back to 1780 Ma using the data from previous studies (Table 3). The overall
range of initial $\varepsilon_{Nd}$ values is from -10.2 ±1.21 to -4.80 ±1.23 (2σ, Fig. 7b). Some
samples from Wafang diorite (namely Muzhijie diorite in some literatures, Ma et al,
2023b; Wang et al, 2016) have enriched Nd isotope composition, which may be

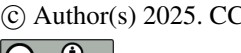



explained by assimilation or contamination of the continental crust due to their higher
zirconium contents (Fig. 7b; Table 3). Overall, the initial $\varepsilon_{Nd}$ values and the
corresponding two-stage Nd model ages ($T_{DM2}$) of the late Paleoproterozoic diorites
are consistent with each other except for the Wafang diorite (Table 3).
The initial $\varepsilon_{Hf}$ values of zircons from the diorites in the NCC have a wide but
consistent range of variations: from -17 to -2.5 in the Gushicun diorite (Ma et al,
2023a; Fig. 7c), from -14 to 0.55 in the Muzhijie diorite (Ma et al, 2023b; Fig. 7c),
and from -17 to 0.95 in the Fudian diorite (Ma et al., 2023b; Fig. 7c). In summary, the
diorites in the NCC have similar Nd-Hf isotopic compositions and form a coherent
group in geochemical diagrams, indicating a close genetic relationship.

**5.2 Initial Sr isotope composition and magma source**
The late Paleoproterozoic diorites in the NCC show a large range in whole-rock initial
Sr isotopic compositions (Fig. 7a; Jiguanshan diorite: 0.7020 to 0.7058; Wafang
diorite: 0.7004 to 0.7050; Shizhaigou diorite: 0.7005 to 0.7053; East-West group dikes:
0.7011 to 0.7053). Determining magma sources for rocks with widely varying initial
Sr ratios is complex, as Sr isotopes can be affected by magma mixing, assimilation,
contamination, and melting degrees. (e.g., Gao et al., 2015; Wolf et al., 2019; Zeng et
al., 2005).
The whole-rock Nd isotopic compositions suggest a heterogeneous magma source
without mixing with the mantle (Fig. 7b). On the other hand, mantle-derived rocks
often have a high MgO content and elevated levels of compatible elements such as Ni
and Cr, which are inconsistent with the elemental content characteristics of these
diorites (Table 1, see previous references).





Variability in Sr isotopic compositions can result from different degrees of source
melting. However, a mica- and feldspar-rich source with high Rb/Sr ratios produces
melts with more radiogenic $^{87}Sr/^{86}Sr$ ratios (e.g., Hu et al., 2018). Melts affected by
the dehydration of amphibole typically have low $^{87}Sr/^{86}Sr$ ratios with adakitic
characteristics (e.g., Rapp and Watson, 1995; Wolf et al., 1993). The different degrees
of source melting are unlikely to be the main cause.
The initial $^{87}Sr/^{86}Sr$ values negatively correlate with the $^{87}Rb/^{86}Sr$ ratios when they are
less than 0.704 (Fig. 7a). When the initial $^{87}Sr/^{86}Sr$ values are greater than 0.704, such
correlation no longer exists. The large uncertainty propagation in calculating the
initial whole-rock Sr isotope compositions for old samples may be the main factor. All
diorites have samples with the initial $^{87}Sr/^{86}Sr$ values greater than 0.704. Excluding
outliers, the mean average initial $^{87}Sr/^{86}Sr$ value is 0.7052 ±0.0003 (2σ, n=8), which
might represent the most likely initial Sr isotopic composition of the source (Fig. 7a).
The initial Sr isotopic compositions of the Xiong'er Group rocks vary widely and tend
to have more variable and radiogenic Sr isotopic ratios (Fig. 7d). The initial Sr isotope
compositions of these diorites are much similar to the lower crustal Archean xenoliths
from the southeastern NCC (initial $^{87}Sr/^{86}Sr$ values: 0.7039–0.7068, t=1780 Ma, e.g.,
Huang et al., 2004), suggesting that they are more likely associated with lower crustal
rocks in the NCC rather than an enriched mantle source like the Xiong'er Group.

**5.3 Petrogenetic considerations**

Several hypotheses have been proposed for the petrogenesis of intermediate dioritic
rocks including partial melting of metasomatized mantle (e.g., Chen et al., 2021),
partial melting of subducted oceanic crust and subsequent melt-peridotite reaction



(e.g., Kelemen, 1995; Stern and Kilian, 1996), magma mixing/mingling (e.g., Reubi
and Blundy, 2009; Streck et al., 2007), melting of basaltic rocks (e.g., Jackson et al.,
2003; Petford and Atherton, 1996), fractional crystallization of basaltic magmas (e.g.,
Castillo et al., 1999).
The diorites from the NCC have similar MgO and low compatible element contents,
suggesting that they were not derived directly from a mantle magma source (Fig. 9a).
The magma mixing/mingling with mantle can also be excluded due to their
homogeneous initial Nd isotope compositions (Fig. 7b), and consistent $SiO_2$ contents
and $Mg^{\#}$ values (Fig. 5d).
Partial melting of the oceanic crust in the subducted slab can also form intermediate
rocks, such as adakites, which often exhibit high Sr/Y ratios (>20) and low Y contents
(<18 ppm) (e.g., Defant and Drummond, 1990; Peacock et al., 1994). The Jiguanshan
and other diorites from the NCC have relatively high Y and Sr contents with Sr/Y
ratios <15. Thus, partial melting of the oceanic crust does not appear to be the reason
for these diorites.
As can be seen from the Harker variation diagrams, the Cr contents decrease with
decreasing MgO, indicating fractionation of clinopyroxene (Fig. 9a). The CaO
contents decrease with increasing $SiO_2$, suggesting crystallization of minerals, such as
plagioclase or clinopyroxene (Fig. 9b). However, $Al_2O_3$ and $Na_2O$ contents do not
significantly decrease with increasing $SiO_2$, indicating that plagioclase and
clinopyroxene were not significant fractionation phases (Fig. 9c-d). The increase in
$K_2O$ contents with increasing $SiO_2$ suggests no biotite and/or K-feldspar fractionation
during magmatic evolution (Fig. 9e). The increasing $SiO_2$ and decreasing $TiO_2$
indicate the crystallization and fractionation of Ti-bearing minerals, such as ilmenite
(Fig 9f). The Eu/Eu$^{*}$ values of the diorites do not show significant changes with Sr



contents, which also proves that fractionation of plagioclase from the melt was not
significant (Fig. 9g). From the above discussion, it can be concluded that the
petrogenesis of the diorites in the NCC was associated with minor fractional
crystallization processes. Whole-rock La/Yb versus La and Zr/Sm versus Zr
correlations are as expected for a partial melting process (Fig. 9h-i). This implies that
the formation of the diorites may be closely related to the partial melting of a basaltic
protolith.
The basement rocks of the lower Taihua Group in the southern margin of the NCC
consist of amphibolite (e.g., Diwu et al., 2014, 2018; Wang et al., 2020). Partial
melting of amphibolite can also lead to the production of intermediate to acidic
magmas (e.g., Beard and Lofgren, 1991; Rapp and Watson, 1995). The amphibolites
of the Taihua Group are characterized by low K content and low $K_2O/Na_2O$ ratios
(<0.5, Wang et al., 2019), making it difficult to generate high-$K_2O$ rocks. (Beard and
Lofgren, 1991; Roberts and Clemens, 1993). The partial melting of amphibolite
typically produces peraluminous melts (e.g., Beard and Lofgren, 1991; Rapp and
Watson, 1995), whereas the diorites in the NCC have low $Al_2O_3$ content with
metaluminous character (Fig. 5c; weight average A/NCK values of 0.81). Additionally,
the $\varepsilon_{Nd}$ values of the Taihua Group amphibolites at t=1780 Ma show a wide range
from -6.7 to 0.4, which is inconsistent with those of the diorites (Wang et al., 2019).
Therefore, it seems unlikely that the diorites formed by the partial melting of Taihua
Group amphibolites.
The mafic rocks in the Xiong'er Group or the mafic dyke swarms are believed to be
the origin of the diorites. (Cui et al., 2011; Ma et al., 2023b; Peng et al., 2007). The
mafic dyke swarms and Xiong'er Group rocks possess a relatively large range of
initial Sr-Nd isotopic compositions (Fig. 7d), while the initial Nd isotopic



compositions of the diorites are relatively homogeneous (Fig. 7b). The whole-rock
initial Nd isotopic compositions and the zircon initial Hf isotope ratios of the Xiong'er
Group rocks are also enriched (Fig. 7c). The initial Pb isotopic compositions of the
mafic dykes and Xiong'er Group rocks are very radiogenic and variable (Fig. 8a, b),
which is due to the high U and Th contents of the protolith, indicating the presence of
an enriched subcontinental lithospheric mantle source (e.g., Hou et al., 2008; Peng et
al., 2004, 2007; Wang et al., 2004, 2010; Zhao et al., 2007). Based on the previous
discussion, the geochemical characteristics of the diorites are more compatible with a
crustal origin. These isotopic compositions of the diorites indicate that their sources
might not have been derived from the enriched mantle.
Additionally, the Xiong'er volcanic rocks have lower Nb/Ta ratios and Nb contents
(Fig. 10a). Nb and Ta share a similar valence state and atomic radii, but they can
undergo fractionation during the subduction process. (Jochum et al., 1986; Shannon,
1976). The Xiong'er volcanic rocks, with higher and positively related Ba/Th and
Sr/Th ratios (Fig. 10a, b), likely originated from a source influenced by early
subduction components, whereas the diorites appear to be less affected by early
subduction-related materials. Therefore, the diorites could be formed by the partial
melting of the mafic protolith of the lower crust on top of an enriched subcontinental
lithospheric mantle beneath the NCC.

**5.4 Tectonic setting**

Diorite is an important intermediate rock that typically forms in island arcs,
subduction zones, and continental collision orogenic belts along the convergent plate
boundaries. Oceanic island arc intermediate rocks are generally characterized by high
MgO, Cr, and Ni contents as boninite and low MgO, high $Al_2O_3$, and $Na_2O/K_2O > 1$



andesite (Hickey et al., 1982; Rapp and Watson, 1995). The continental arc
intermediate rocks typically show high $Al_2O_3$ content with a wider range of $^{87}Sr/^{86}Sr$
and $^{143}Nd/^{144}Nd$ isotope compositions, reflecting an obvious influence of continental
crust more complex and enriched source (Hawkesworth et al., 1979; Peacock et al.,
1994). The Paleoproterozoic diorites in the NCC lack these features of arc-related
rocks, meanwhile, their trace element distributions differ from those of island arc and
continental arc intermediate rocks. For example, these diorites do not have significant
enrichment in Sr, Th, and U in the primitive mantle-normalized diagram as arc-related
rocks (Fig. 6a). These diorites also exhibit a negative Eu anomaly in the REE diagram,
which is different from the arc-related rocks (Fig. 6b). The diorites in collisional
orogenic belts have high MgO and $K_2O$ contents and adakite-like characteristics with
high Sr/Y and La/Yb ratios (Yang et al., 2015). However, Paleoproterozoic diorites in
the North China Craton do not show the typical arc-related element and isotopic
signatures, suggesting a different formation environment from subduction-related
magmatism.
Diorites can still form through crustal extension (Asmerom et al., 1990; Liu et al.,
2024). The North China Craton was in a post-collisional extensional environment
after the amalgamation (Zhai, 2010), where the magmatic genesis became more
complex (Bonin, 2004). Zircon is relatively stable and may record more information,
therefore, its trace elements offer significant potential for distinguishing between
different tectonic environments. Zircon samples with La contents less than 1 ppm
were selected for discussion to ensure accurate information from zircon trace element
contents without interference from the inclusion of other accessory phases (Zou et al.,
2019). All zircons from the diorites plot within the continental area in the U/Yb versus
Y diagram (Fig. 11a), and most of them tend to fall into a rift-controlled tectonic



environment in the zircon tectonic discrimination diagrams (Fig. 11b, c; Carly et al.,

407 2014).

Furthermore, high-field strength elements, such as Zr, Nb, Ta, Hf, and Th, are
important in tectonic discrimination diagrams. The distinctive Th content in arc
magmas is primarily due to its low solubility in subduction zone fluids and its
contribution from sedimentary components (e.g., Bailey and Ragnasdottir, 1994;
Pearce and Peate, 1995). The arc-related/orogenic magmas usually have less Nb than
those in within-plate settings (e.g., Pearce and Peate, 1995; Sun and McDonough,
1989). Nb in zircon is thought to be incorporated through xenotime-type substitution
(Schulz et al., 2006) and is suggested to reflect the magma composition with minimal
influence from magmatic fractionation (Hoskin et al., 2000; Schulz et al., 2006). In
the Nb/Hf versus Th/U and Hf/Th versus Th/Nb diagrams, zircons from the Fudian
and Gushicun diorites plot both within the arc-related/orogenic area and near this area
(Fig. 11d, e). The Jiuganshan and Muzhijie diorites plot both in the
arc-related/orogenic and within-plate/anorogenic areas (Fig. 11d, e). The whole-rock
Ta/Yb and Th/Yb ratios of these diorites are uniform (Fig. 11f), all falling within the
overlapping area of the ACM (active continental margins) and WPVZ (Within-Plate
Volcanic Zone). These may indicate that the post-collisional extension during this
period may ultimately lead to rift evolution continuously and progressively. The
diorites preserve a record of the superimposition of representative components from
multiple tectonic settings.
After the 1.85 Ga collisional event, the North China Craton entered a prolonged
post-collisional extensional stage. The magmatism was primarily controlled by crustal
remelting, leading to the widespread formation of various crust-derived granites in the
orogenic belts at the end of the Paleoproterozoic (Deng et al., 2016; Wang et al.,



2023b; Xu et al., 2024). However, after 1.78 Ga, the crust-derived diorites show
transitional features in their tectonic setting, retaining some remnant effects of the
orogenic magmatism while gradually evolving toward intraplate magmatism. It
reflects the ongoing extension of the North China Craton after its amalgamation.

**6 Conclusions**
The Jiguanshan diorite yields a zircon U-Pb age of about 1.78 Ga. It displays
geochemical features in common with other diorite intrusions within the NCC. The
diorite intrusion was contemporaneous with the Xiong'er volcanic rocks and the mafic
dyke swarms, representing a significant period of magmatism.
The late Paleoproterozoic diorites primarily resulted from the partial melting of the
mafic protolith. The Sr-Nd-Pb-Hf isotopic characteristics indicate that the source was
not the same as that for the Xiong'er volcanic rocks or mafic dyke swarms. Instead,
they are more likely derived from the lower crust of the NCC.
The formation of Paleoproterozoic diorites in the North China Craton is unlikely to be
arc-related. Instead, it is associated with a rift setting. The formation of diorite records
the transition of crustal origin rocks from orogenic-related magmatism to intraplate
magmatism during the post-collision extensional stage. It reflects the ongoing
extension of the North China Craton after its amalgamation.

**Acknowledgements**
This study was financially supported by the Strategic Priority Research Program of
the Chinese Academy of Sciences (grant Nos. XDA0430203) and the National
Natural Science Foundation of China (grant Nos. 42202069 and 41872049). Zhiyi



Wang was financially supported by China Scholarship Council (202306340065). We
thank P. Xiao and Z.-H. Hou for assistance with the analysis.

**Author contributions:**
Zhiyi Wang: Investigation, Writing - Review & Editing;
Shuhao Tang, Yiru Ji, Jianfeng He: Investigation, Review & Editing;
Wolfgang Siebel.: Conceptualization, Writing - Review & Editing;
Jun He & Fukun Chen: Supervision, Writing - Review & Editing, Funding acquisition.

**Competing interests:**
The authors declare no competing financial and non-financial interests for this study.




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



**Figures**

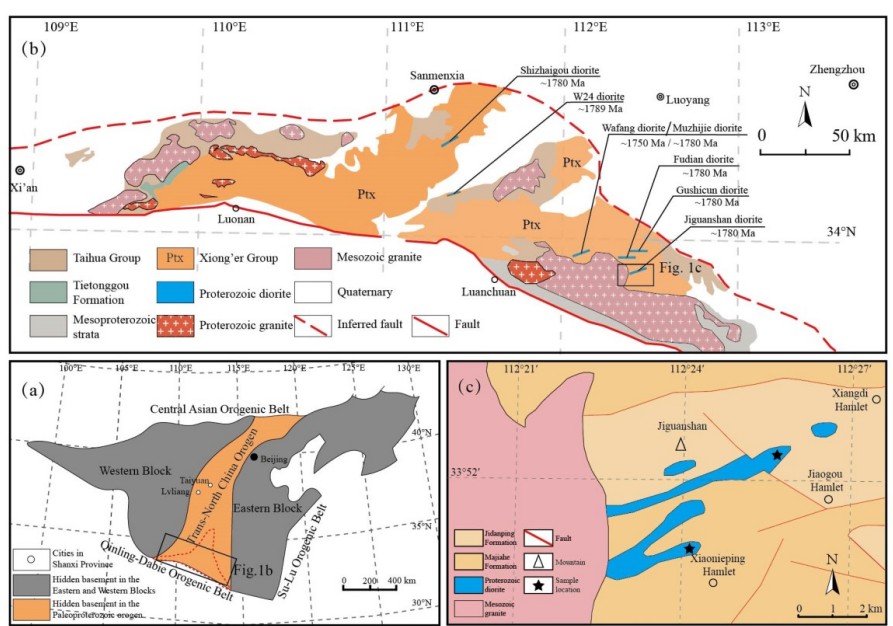


**Figure 1** (a) Tectonic sketch of the North China Craton (after Zhao et al., 2001); (b) Geological map of the southern margin of the North China Craton (after Diwu et al., 2014; diorites from Cui et al., 2011; Ma et al 2023a, b; Wang et al., 2016; Zhao et al., 2004); (c) Geological map of the Jiguanshan diorite (after BGMRH, 1994)

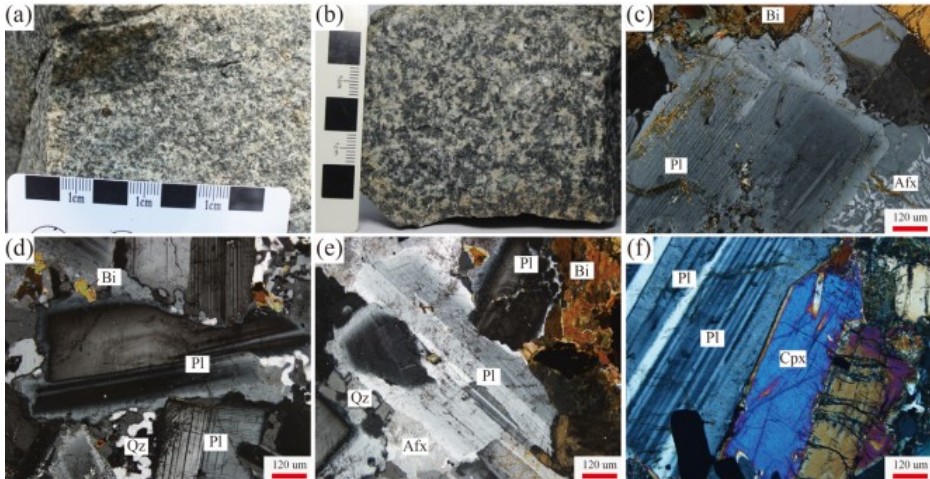



**Figure 2** (a-b) Field photographs and representative hand specimens of the Jiguanshan diorite; (c-f)
Micrographs under the plane-polarized light of the Jiguanshan diorite. Mineral abbreviations: Afs,
alkali feldspar; Bi, biotite; Cpx, Clinopyroxene; Pl, plagioclase; Qz, quartz

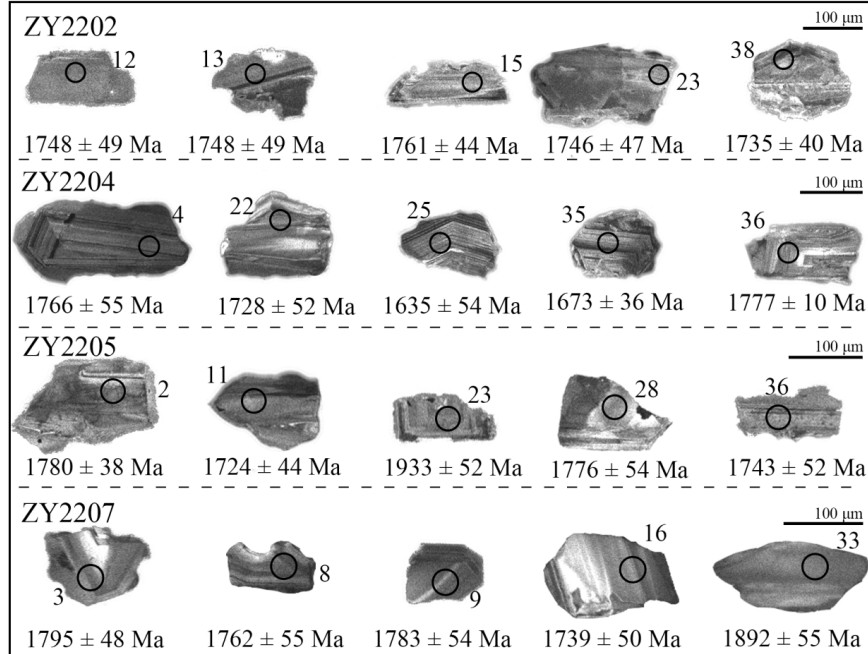

**Figure 3** Cathodoluminescence (CL) images of representative zircon grains from the Jiguanshan
diorite





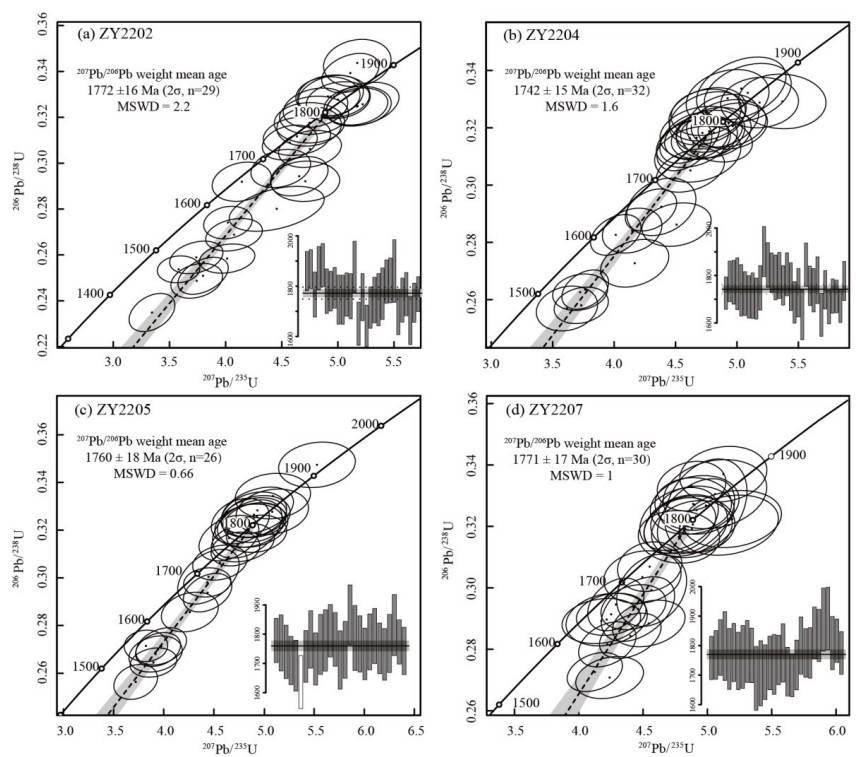

**Figure 4** (a-d) Zircon U–Pb concordia diagrams of the Jiguanshan diorite



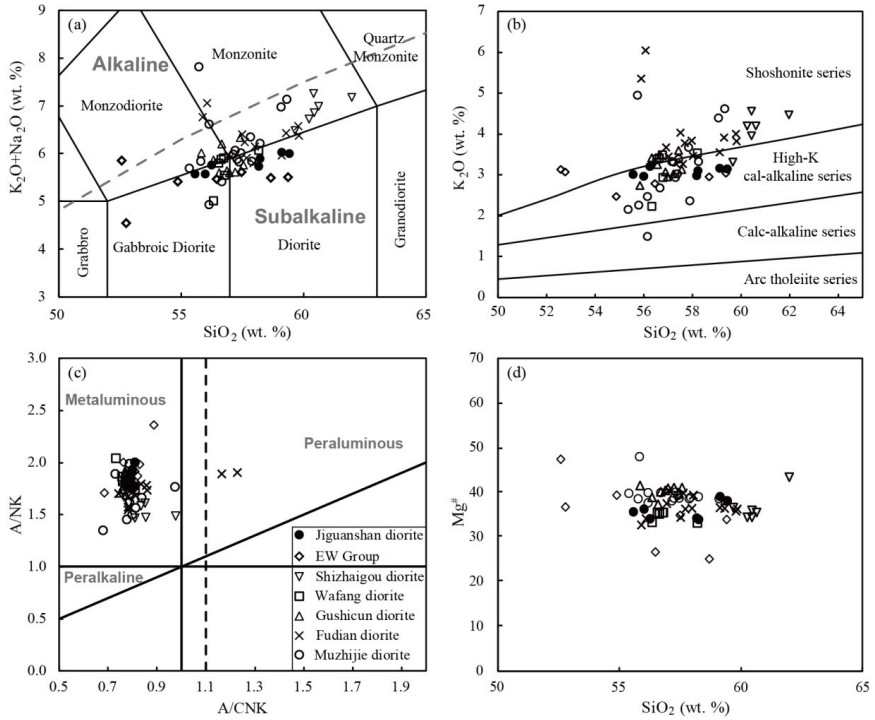

Figure 5 Plots of major elements for the diorites: (a) TAS diagram (after Le Bas et al., 1986); (b) K$_2$O content versus SiO$_2$ content (after Peccerillo and Taylor, 1976); (c) A/NK versus A/CNK values (after Maniar and Piccoli, 1989) (d) Mg$^\#$ value versus SiO$_2$ content (wt. %)

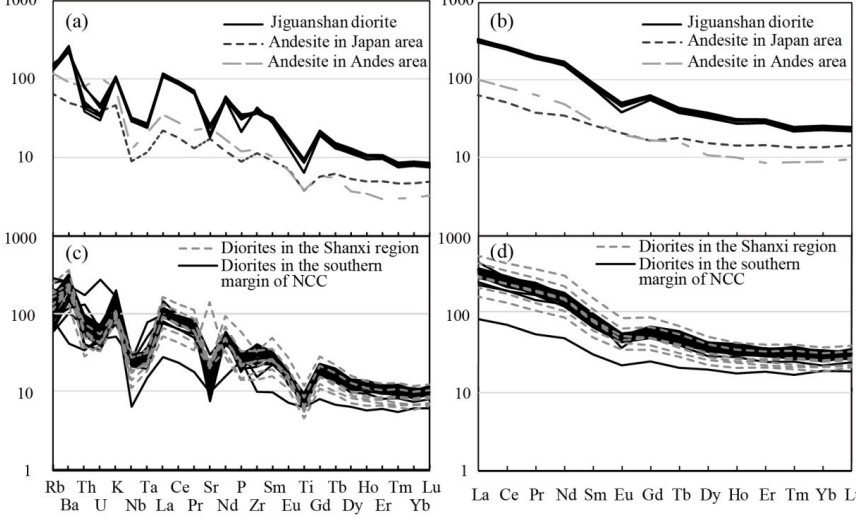



**Figure 6** Primitive-mantle normalized trace element spider diagrams and chondrite-normalized
REE patterns of diorites. Normalization values from Sun and McDonough (1989); Diorites in
Shanxi region (Peng et al., 2007), Diorites in the southern margin of the NCC (Cui et al., 2011;
Ma et al 2023a, b; Wang et al., 2016; Zhao et al., 2004). The average trace element
compositions of intermediate rocks in the Japan arc and Andes arc are derived from Pan et al.
(2017).


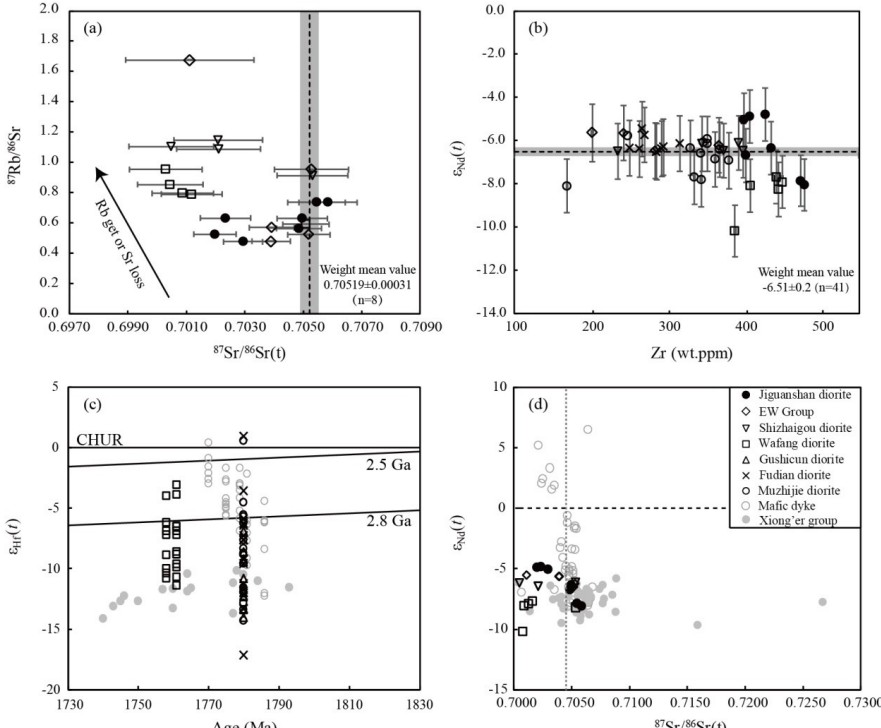

**Figure 7** (a) $^{87}$Rb/$^{86}$Sr value versus $^{87}$Sr/$^{86}$Sr(t) value; (b) $\varepsilon_{Nd}$(t) value versus Zr content (ppm); (c)
$\varepsilon_{Nd}$(t) value versus age (Ma); (d) $\varepsilon_{Nd}$(t) value versus $^{87}$Sr/$^{86}$Sr(t) value. Data source of the
Xiong'er Group (Hf isotope composition, Wang et al., 2010; initial Sr isotope composition
and initial $\varepsilon_{Nd}$ value, He et al., 2008, 2010; Peng et al., 2008; Wang et al., 2010; Zhao et al.,
2002); mafic dyke swarms (initial Sr isotope composition and initial $\varepsilon_{Nd}$ value, Hu et al.,
2010; Peng et al., 2007; Wang et al., 2004)




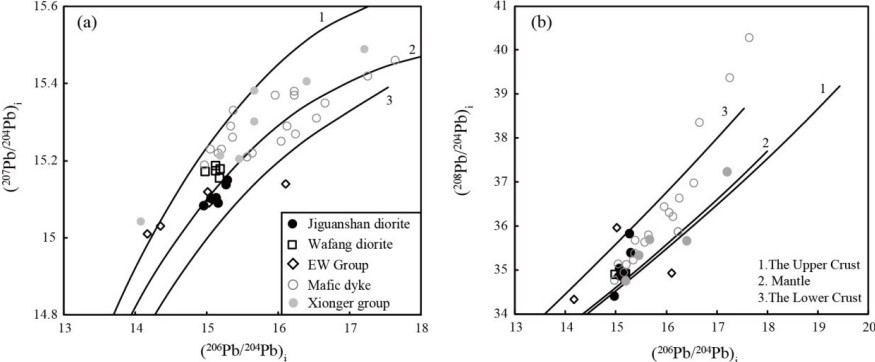


**Figure 8** (a) ($^{207}Pb/^{204}Pb$)i versus ($^{206}Pb/^{204}Pb$)i; (b) ($^{208}Pb/^{204}Pb$)i versus ($^{206}Pb/^{204}Pb$)i. The data
source of the Xiong'er Group (Pb isotope composition, Zhao, 2000); mafic dyke swarms
(initial Pb isotope composition, Hu et al., 2010; Peng et al., 2007); diorites (initial Pb isotope
composition, Peng et al., 2007; Wang et al., 2016)

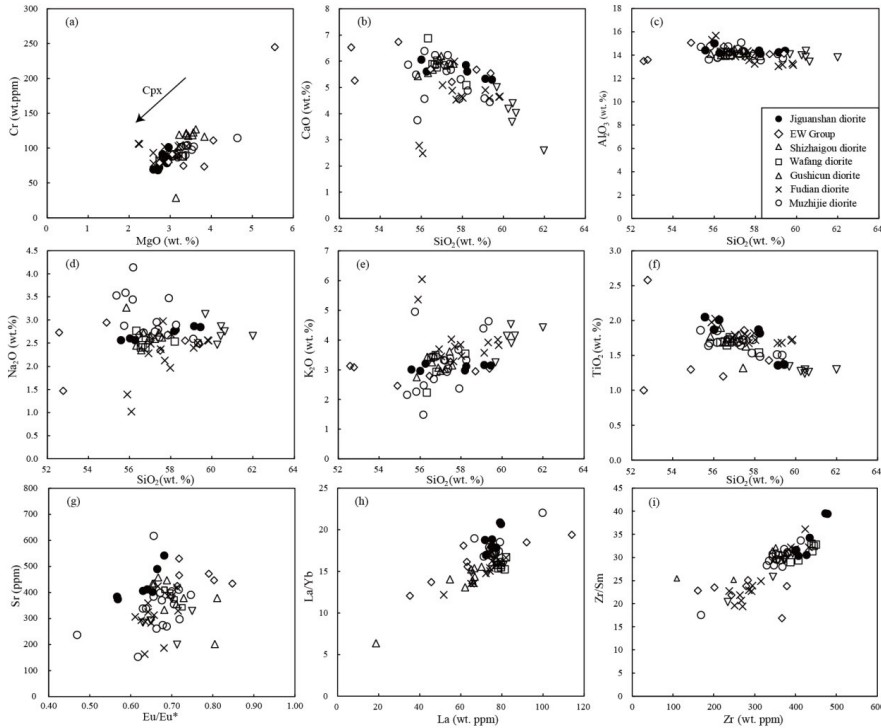

**Figure 9** (a) Cr (ppm) content versus MgO content (wt. %); (b) CaO (wt. %) content versus SiO$_2$
content (wt. %); (c) Al$_2$O$_3$ (wt. %) content versus SiO$_2$ content (wt. %); (d) Na$_2$O (wt. %)
content versus SiO$_2$ content (wt. %); (e) K$_2$O (wt. %) content versus SiO$_2$ content (wt. %); (f)





TiO$_2$ (wt. %) content versus SiO$_2$ content (wt. %);    (g) Eu/Eu*value versus Sr content (ppm);

(h) La/Yb value versus La content (ppm); (i) Zr/Sm value versus Zr content (ppm)

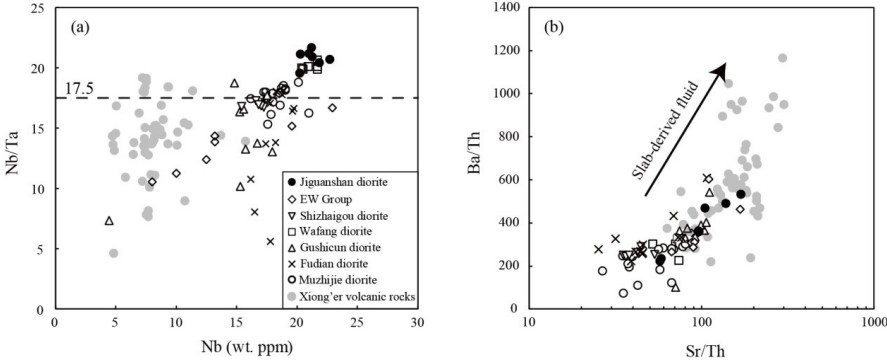

**Figure 10** (a) Nb/Ta versus Nb content (ppm); (b) Ba/Th value versus Sr/Th values; Data source

of the Xiong'er Group (He et al., 2008, 2010; Wang et al., 2010; Zhao et al., 2002)

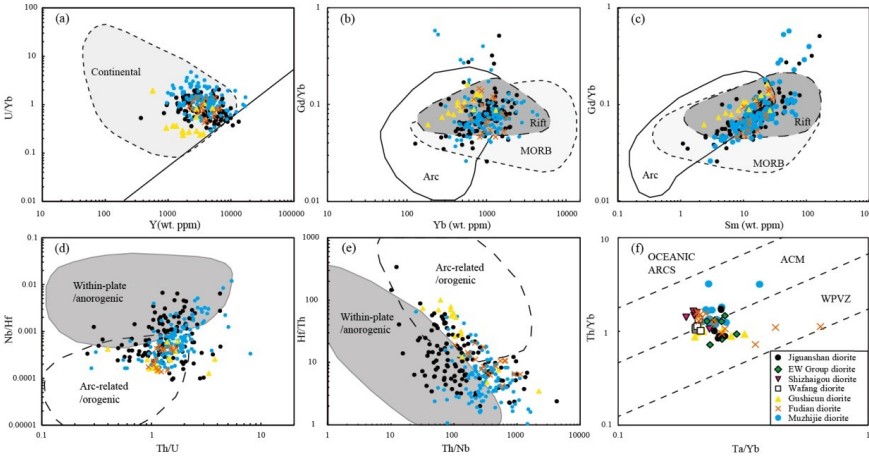

**Figure 11** (a) Zircon trace element U/Yb value versus Y (ppm) (after Grimes et al., 2007); (b)

Zircon trace element Gd/Yb value versus Yb (ppm) (after Carley et al., 2014); (c)   Zircon

trace element Gd/Yb value versus Sm (ppm) (after Carley et al., 2014); (d) Zircon trace

element Nb/Hf value versus Th/U value (after Hawkesworth and Kemp, 2006); (e) Zircon

trace element Hf/Th value versus Th/Nb value (after Yang et al., 2012); (f) Whole-rock trace

element Th/Yb value versus Ta/Yb value (after Pearce, 1983; Gorton and Schandl, 2000);





**Tables**

**Table 1** Major (wt. %) and trace element contents (ppm) of the Jiguanshan diorite

| Sample No. | ZY2201 | ZY2202 | ZY2203 | ZY2204 | ZY2205 | ZY2206 | ZY2207 |
|---|---|---|---|---|---|---|---|
| (wt.%) | | | | | | | |
| $SiO_2$ | 58.18 | 59.44 | 59.13 | 58.24 | 56.26 | 56.01 | 55.57 |
| $TiO_2$ | 1.87 | 1.37 | 1.36 | 1.82 | 2.01 | 1.87 | 2.05 |
| $Al_2O_3$ | 14.38 | 14.37 | 14.24 | 14.11 | 14.18 | 15.00 | 14.41 |
| $^TFe_2O_3$ | 10.38 | 9.04 | 9.17 | 10.00 | 10.35 | 10.18 | 10.50 |
| MnO | 0.15 | 0.14 | 0.14 | 0.14 | 0.17 | 0.14 | 0.15 |
| MgO | 2.73 | 2.81 | 2.96 | 2.59 | 2.70 | 2.92 | 2.94 |
| CaO | 5.85 | 5.29 | 5.33 | 5.60 | 5.61 | 6.06 | 5.81 |
| $Na_2O$ | 2.76 | 2.85 | 2.87 | 2.79 | 2.56 | 2.60 | 2.56 |
| $K_2O$ | 2.98 | 3.15 | 3.16 | 3.11 | 3.21 | 2.97 | 3.01 |
| $P_2O_5$ | 0.71 | 0.46 | 0.45 | 0.65 | 0.73 | 0.68 | 0.76 |
| LOI | 0.48 | 1.31 | 0.67 | 0.36 | 1.53 | 1.60 | 1.67 |
| Total | 100.47 | 100.23 | 99.48 | 99.41 | 99.31 | 100.03 | 99.43 |
| (ppm) | | | | | | | |
| Li | 11.2 | 19.8 | 19.9 | 14.8 | 18.6 | 20.7 | 18.2 |
| Be | 2.66 | 2.80 | 2.76 | 2.94 | 3.06 | 2.70 | 2.97 |
| Sc | 22.7 | 20.1 | 20.4 | 23.3 | 24.3 | 24.0 | 23.8 |
| V | 163 | 141 | 147 | 168 | 179 | 165 | 164 |
| Cr | 72.1 | 91.3 | 101.3 | 69.5 | 68.6 | 78.6 | 83.5 |
| Ni | 21.3 | 22.3 | 24.0 | 20.7 | 19.2 | 20.2 | 21.6 |
| Cu | 20.8 | 19.8 | 19.9 | 20.9 | 27.0 | 22.2 | 23.3 |
| Zn | 131 | 128 | 122 | 133 | 148 | 139 | 141 |
| Ga | 21.9 | 21.9 | 21.8 | 22.9 | 23.3 | 23.8 | 22.7 |
| Rb | 80.3 | 95.2 | 97.8 | 88.4 | 88.0 | 89.5 | 88.9 |
| Sr | 412 | 374 | 384 | 406 | 403 | 542 | 490 |
| Y | 47.5 | 44.4 | 43.8 | 48.4 | 49.3 | 44.8 | 46.7 |
| Zr | 402 | 478 | 474 | 435 | 428 | 400 | 407 |
| Nb | 20.2 | 21.2 | 21.0 | 21.2 | 22.7 | 20.3 | 21.8 |
| Cs | 0.60 | 0.77 | 0.74 | 0.95 | 2.98 | 3.63 | 4.44 |
| Ba | 1543 | 1515 | 1504 | 1544 | 1814 | 1714 | 1737 |
| La | 72.2 | 79.0 | 79.5 | 75.0 | 77.3 | 71.7 | 75.2 |
| Ce | 149 | 161 | 161 | 154 | 163 | 150 | 159 |
| Pr | 17.6 | 18.3 | 18.1 | 18.2 | 19.4 | 18.0 | 18.9 |
| Nd | 72.3 | 71.2 | 70.9 | 73.2 | 80.0 | 72.9 | 77.1 |
| Sm | 12.7 | 12.1 | 12.0 | 12.7 | 14.0 | 12.8 | 13.4 |
| Eu | 2.63 | 2.21 | 2.18 | 2.59 | 2.93 | 2.78 | 2.87 |
| Gd | 12.1 | 11.2 | 11.2 | 12.1 | 13.0 | 11.7 | 12.5 |
| Tb | 1.53 | 1.39 | 1.40 | 1.51 | 1.63 | 1.47 | 1.56 |
| Dy | 8.99 | 8.32 | 8.11 | 8.92 | 9.50 | 8.53 | 9.00 |
| Ho | 1.67 | 1.54 | 1.53 | 1.67 | 1.75 | 1.53 | 1.65 |
| Er | 4.97 | 4.56 | 4.54 | 4.95 | 5.09 | 4.55 | 4.87 |
| Tm | 0.62 | 0.55 | 0.55 | 0.60 | 0.63 | 0.55 | 0.58 |



| | | | | | | | |
|---|---|---|---|---|---|---|---|
| Yb | 4.26 | 3.79 | 3.84 | 4.18 | 4.33 | 3.82 | 3.99 |
| Lu | 0.61 | 0.55 | 0.56 | 0.60 | 0.63 | 0.55 | 0.58 |
| Hf | 7.97 | 9.09 | 9.15 | 8.20 | 8.46 | 7.59 | 7.98 |
| Ta | 1.03 | 0.98 | 0.99 | 1.01 | 1.10 | 0.96 | 1.07 |
| Pb | 16.4 | 21.2 | 18.0 | 16.3 | 18.9 | 15.2 | 14.2 |
| Th | 4.28 | 6.43 | 6.71 | 4.27 | 3.87 | 3.22 | 3.55 |
| U | 0.70 | 0.98 | 0.88 | 0.71 | 0.75 | 0.61 | 0.68 |
| | | | | | | | |
| $K_2O/Na_2O$ | 1.08 | 1.11 | 1.10 | 1.11 | 1.25 | 1.14 | 1.18 |
| $K_2O+Na_2O$ (Wt.%) | 5.74 | 6.00 | 6.03 | 5.90 | 5.77 | 5.57 | 5.57 |
| Mg# | 34.5 | 38.3 | 39.2 | 34.1 | 34.3 | 36.5 | 35.9 |
| A/CNK | 0.78 | 0.81 | 0.80 | 0.78 | 0.79 | 0.81 | 0.80 |
| A/NK | 1.85 | 1.77 | 1.75 | 1.77 | 1.84 | 2.00 | 1.93 |
| ΣREE | 361.5 | 375.8 | 375.1 | 370.4 | 393.2 | 361.2 | 381.3 |
| Eu/Eu* | 0.64 | 0.57 | 0.57 | 0.63 | 0.65 | 0.68 | 0.66 |
| $(La/Yb)_N$ | 12.2 | 15.0 | 14.8 | 12.9 | 12.8 | 13.5 | 13.5 |

$Mg^{\#}=(MgO+FeO_{total})/MgO\times100$
$Eu/Eu^{*}=2Eu_N/(Sm_N+Gd_N)$; $(La/Yb)_N$=chondrite-normalized La/Yb ratio



**Table 2** Whole-rock Sr isotopic compositions of the late Paleoproterozoic diorites in the NCC

| Sample | Age (Ma) | Rb (ppm) | Sr (ppm) | Rb/Sr | $^{87}Rb/^{86}Sr$ | $^{87}Sr/^{86}Sr$ | ±2SE | $^{87}Sr/^{86}Sr$ (t) | Error (abs.) | Data source |
|---|---|---|---|---|---|---|---|---|---|---|
| Jiguanshan diorite | | | | | | | | | | |
| ZY2201 | 1780 | 80.3 | 412 | 0.20 | 0.5648 | 0.71931 | 0.000010 | 0.70485 | 0.00077 | This study |
| ZY2202 | 1780 | 95.2 | 374 | 0.25 | 0.7371 | 0.72471 | 0.000012 | 0.70584 | 0.00099 | |
| ZY2203 | 1780 | 97.8 | 384 | 0.25 | 0.7377 | 0.72434 | 0.000011 | 0.70546 | 0.00099 | |
| ZY2204 | 1780 | 88.4 | 406 | 0.22 | 0.6307 | 0.72111 | 0.000011 | 0.70496 | 0.00085 | |
| ZY2205 | 1780 | 88.0 | 403 | 0.22 | 0.6334 | 0.71856 | 0.000011 | 0.70235 | 0.00086 | |
| ZY2206 | 1780 | 89.5 | 542 | 0.17 | 0.4780 | 0.71518 | 0.000011 | 0.70294 | 0.00066 | |
| ZY2207 | 1780 | 88.9 | 490 | 0.18 | 0.5252 | 0.71542 | 0.000013 | 0.70198 | 0.00072 | |
| Wafang diorote | | | | | | | | | | |
| WF1307-3 | 1780 | 107.0 | 389 | 0.28 | 0.7969 | 0.72131 | 0.000013 | 0.70091 | 0.00106 | Wang et al. (2016) |
| WF1307-4 | 1780 | 109.0 | 400 | 0.27 | 0.7895 | 0.72144 | 0.000014 | 0.70123 | 0.00105 | |
| WF1307-5 | 1780 | 84.0 | 411 | 0.20 | 0.5921 | 0.72024 | 0.000016 | 0.70508 | 0.00080 | |
| WF1307-8 | 1780 | 113.0 | 343 | 0.33 | 0.9548 | 0.72479 | 0.000016 | 0.70035 | 0.00127 | |
| WF1307-9 | 1780 | 110.0 | 373 | 0.29 | 0.8545 | 0.72236 | 0.000014 | 0.70048 | 0.00114 | |
| Shizhaigou diorite | | | | | | | | | | |
| Ln-1 | 1780 | 103.7 | 272 | 0.38 | 1.1040 | 0.72874 | 0.000012 | 0.70048 | 0.00146 | Cui et al. (2011) |
| Ln-2 | 1780 | 101.5 | 322 | 0.31 | 0.9125 | 0.72868 | 0.000015 | 0.70532 | 0.00121 | |
| Ln-3 | 1780 | 136.4 | 200 | 0.68 | 1.9758 | 0.72509 | 0.00001 | 0.67452 | 0.00259 | |
| Ln-4 | 1780 | 116.6 | 295 | 0.40 | 1.1479 | 0.73149 | 0.000015 | 0.70210 | 0.00152 | |
| Ln-5 | 1780 | 112.5 | 300 | 0.38 | 1.0885 | 0.72997 | 0.000014 | 0.70211 | 0.00144 | |
| E-W Group dyke | | | | | | | | | | |
| 02SX001 | 1780 | 154.8 | 470 | 0.33 | 0.9542 | 0.72970 | 0.000014 | 0.70528 | 0.00127 | Peng et al. (2007) |
| 02SX007 | 1780 | 81.2 | 450 | 0.18 | 0.5231 | 0.71858 | 0.000014 | 0.70519 | 0.00072 | |
| 03LF01 | 1780 | 74.4 | 449 | 0.17 | 0.4801 | 0.71619 | 0.000013 | 0.70390 | 0.00066 | |
| 03FS04 | 1780 | 131.8 | 229 | 0.58 | 1.6748 | 0.74399 | 0.000012 | 0.70112 | 0.00 |



| 03FS07 | 1780 | 106.0 | 539 | 0.20 | 0.5699 | 0.71852 | 0.000013 | 0.70393 | 0.00078 | 220 |
| Weight mean value | | | | | | | | 0.70519 | 0.00031 | (n=8, calculated by IsoplotR) |

$(^{87}Sr/^{86}Sr)_s = (^{87}Sr/^{86}Sr)_0 + (^{87}Rb/^{86}Sr)_s \times (e^{\lambda t} - 1)$
$\lambda_{87Rb} = 1.42 \times 10^{-11}/a^{-1}$
Error of initial ratio is calculated from the measurement error of the isotope ratio, the estimated
concentration error and the age error. The decay constant is considered to be a fixed value.
$\sigma_{Sr(t)}$ is mean-square deviation of $(^{87}Sr/^{86}Sr)_t$
$\sigma_{Rb}$ is mean-square deviation of $(^{87}Rb/^{86}Sr)_s$
$\sigma_t$ is mean-square deviation of age
$$\sigma_{Sr(t)} = \sqrt{\sigma_{Sr}^2 + \sigma_{Rb}^2(e^{\lambda t} - 1)^2 + \sigma_t^2(\lambda e^{\lambda t}(\frac{87_{Rb}}{86_{Sr}}))^2}$$





**Table 3** Whole-rock Nd isotopic compositions of the late Paleoproterozoic diorites in the NCC

| Sample | Age (Ma) | Nd (ppm) | Sm (ppm) | $^{147}Sm/^{144}Nd$ | $^{143}Nd/^{144}Nd$ | Error (2s) | $^{143}Nd/^{144}Nd(t)$ |
|---|---|---|---|---|---|---|---|
| **Jiguanshan diorite** | | | | | | | |
| ZY2201 | 1780 | 72.3 | 12.7 | 0.1063 | 0.511238 | 0.000007 | 0.509994 |
| ZY2202 | 1780 | 71.2 | 12.1 | 0.1029 | 0.511129 | 0.000008 | 0.509924 |
| ZY2203 | 1780 | 70.9 | 12.0 | 0.1022 | 0.511131 | 0.000005 | 0.509934 |
| ZY2204 | 1780 | 73.2 | 12.7 | 0.1049 | 0.511240 | 0.000007 | 0.510011 |
| ZY2205 | 1780 | 80.0 | 14.0 | 0.1058 | 0.511329 | 0.000007 | 0.510090 |
| ZY2206 | 1780 | 72.9 | 12.8 | 0.1058 | 0.511317 | 0.000005 | 0.510078 |
| ZY2207 | 1780 | 77.1 | 13.4 | 0.1054 | 0.511320 | 0.000006 | 0.510086 |
| **E-W Group dyke** | | | | | | | |
| 02SX001 | 1780 | 113 | 20.3 | 0.1084 | 0.511287 | 0.000009 | 0.510018 |
| 02SX007 | 1780 | 62.6 | 11.3 | 0.1093 | 0.511285 | 0.000010 | 0.510005 |
| 03LF01 | 1780 | 45.1 | 8.36 | 0.1120 | 0.511358 | 0.000017 | 0.510047 |
| 03FS04 | 1780 | 102 | 17.5 | 0.1039 | 0.511270 | 0.000010 | 0.510053 |
| 03FS07 | 1780 | 62.7 | 11.1 | 0.1068 | 0.511297 | 0.000013 | 0.510047 |
| **Shizhaigou diorite** | | | | | | | |
| Ln-1 | 1780 | 69.0 | 12.3 | 0.1075 | 0.511280 | 0.000012 | 0.510021 |
| Ln-2 | 1780 | 66.4 | 11.7 | 0.1065 | 0.511270 | 0.000011 | 0.510023 |
| Ln-3 | 1780 | 61.9 | 11.2 | 0.1090 | 0.511280 | 0.000011 | 0.510003 |
| Ln-4 | 1780 | 71.1 | 12.6 | 0.1072 | 0.511260 | 0.000011 | 0.510005 |
| Ln-5 | 1780 | 69.4 | 12.3 | 0.1072 | 0.511260 | 0.000012 | 0.510005 |
| **Wafang diorote** | | | | | | | |
| WF1307-3 | 1780 | 78.4 | 13.7 | 0.1056 | 0.511169 | 0.000008 | 0.509953 |
| WF1307-4 | 1780 | 78.5 | 14.1 | 0.1086 | 0.511215 | 0.000008 | 0.509965 |
| WF1307-5 | 1780 | 75.9 | 13.7 | 0.1091 | 0.511192 | 0.000008 | 0.509936 |
| WF1307-8 | 1780 | 77.6 | 13.4 | 0.1044 | 0.511039 | 0.000007 | 0.509837 |
| WF1307-9 | 1780 | 77.5 | 13.9 | 0.1084 | 0.511193 | 0.000005 | 0.509945 |
| **Gushicun diorite** | | | | | | | |
| 20XRδ-1 | 1780 | 58.0 | 10.9 | 0.1134 | 0.511327 | 0.000004 | 0.509999 |
| 20XRδ-3 | 1780 | 63.3 | 11.7 | 0.1118 | 0.511334 | 0.000006 | 0.510025 |
| 20XRδ-4 | 1780 | 59.1 | 10.9 | 0.1118 | 0.511341 | 0.000006 | 0.510032 |
| 20XRδ-5 | 1780 | 53.1 | 9.9 | 0.1122 | 0.511354 | 0.000006 | 0.510041 |



| The Muzhijie diorites | | | | | | | |
|---|---|---|---|---|---|---|---|
| 20δPt2-1 | 1780 | 63.5 | 11.5 | 0.1090 | 0.511297 | 0.000004 | 0.510021 |
| 20δPt2-3 | 1780 | 64.2 | 11.7 | 0.1100 | 0.511300 | 0.000004 | 0.510012 |
| 20δPt2-5 | 1780 | 66.4 | 12.3 | 0.1122 | 0.511295 | 0.000007 | 0.509982 |
| 20δPt2-7 | 1780 | 72.1 | 13.1 | 0.1101 | 0.511297 | 0.000008 | 0.510007 |
| 20δPt2-9 | 1780 | 54.2 | 9.6 | 0.1076 | 0.511181 | 0.000006 | 0.509922 |
| 20δPt2-11 | 1780 | 64.5 | 11.4 | 0.1073 | 0.511199 | 0.000006 | 0.509943 |
| 20δPt2-13 | 1780 | 62.9 | 11.2 | 0.1076 | 0.511196 | 0.000008 | 0.509937 |
| 20δPt2-16 | 1780 | 67.9 | 12.3 | 0.1098 | 0.511270 | 0.000007 | 0.509984 |
| | | | | | | | |
| Fudian diorite | | | | | | | |
| 20XRSC-1 | 1780 | 65.8 | 12.1 | 0.1110 | 0.511309 | 0.000006 | 0.510009 |
| 20XRSC-2 | 1780 | 67.1 | 12.3 | 0.1111 | 0.511315 | 0.000006 | 0.510014 |
| 20XRSC-3 | 1780 | 69.5 | 12.8 | 0.1113 | 0.511314 | 0.000004 | 0.510011 |
| 20XRSC-4 | 1780 | 67.5 | 12.5 | 0.1117 | 0.511311 | 0.000007 | 0.510002 |
| 20XRSC-5 | 1780 | 70.1 | 12.9 | 0.1111 | 0.511311 | 0.000006 | 0.510010 |
| 20XRSC-6 | 1780 | 68.9 | 12.7 | 0.1112 | 0.511324 | 0.000005 | 0.510022 |
| 20XRSC-8 | 1780 | 71.7 | 12.9 | 0.1089 | 0.511331 | 0.000006 | 0.510056 |
| 20XRSC-9 | 1780 | 76.6 | 13.9 | 0.1096 | 0.511325 | 0.000005 | 0.510042 |
| | | | | | | | |
| Weight mean value | | | | | | | |







| Error (abs.) | $\varepsilon_{Nd}(t)$ | Error ($\varepsilon Nd$) | $T_{DM2}$ (Ga) | Data source |
|---|---|---|---|---|
| 0.000063 | -6.69 | 1.24 | 2.83 | |
| 0.000061 | -8.04 | 1.20 | 2.94 | |
| 0.000060 | -7.85 | 1.19 | 2.93 | |
| 0.000062 | -6.35 | 1.22 | 2.80 | This study |
| 0.000063 | -4.80 | 1.23 | 2.68 | |
| 0.000063 | -5.03 | 1.23 | 2.70 | |
| 0.000062 | -4.88 | 1.22 | 2.68 | |
| | | | | |
| 0.000065 | -6.21 | 1.27 | 2.79 | |
| 0.000065 | -6.47 | 1.28 | 2.81 | |
| 0.000068 | -5.64 | 1.34 | 2.75 | Peng et al. (2007) |
| 0.000062 | -5.53 | 1.22 | 2.74 | |
| 0.000064 | -5.65 | 1.26 | 2.75 | |
| | | | | |
| 0.000065 | -6.15 | 1.26 | 2.79 | |
| 0.000064 | -6.10 | 1.25 | 2.78 | |
| 0.000065 | -6.50 | 1.28 | 2.82 | Cui et al. (2011) |
| 0.000064 | -6.46 | 1.26 | 2.81 | |
| 0.000064 | -6.46 | 1.26 | 2.81 | |
| | | | | |
| 0.000062 | -7.90 | 1.23 | 2.93 | |
| 0.000063 | -7.67 | 1.26 | 2.91 | |
| 0.000064 | -8.24 | 1.27 | 2.96 | Wang et al. (2016) |
| 0.000061 | -10.2 | 1.21 | 3.11 | |
| 0.000063 | -8.07 | 1.26 | 2.94 | |
| | | | | |
| 0.000067 | -6.58 | 1.31 | 2.82 | |
| 0.000066 | -6.08 | 1.30 | 2.78 | |
| 0.000066 | -5.94 | 1.30 | 2.77 | Ma et al. (2023a) |
| 0.000066 | -5.77 | 1.30 | 2.76 | |
| | | | | |
| 0.000064 | -6.15 | 1.26 | 2.79 | Ma et al. (2023b) |

| | | | |
|---|---|---|---|
| 0.000065 | -6.33 | 1.27 | 2.80 |
| 0.000067 | -6.92 | 1.30 | 2.85 |
| 0.000065 | -6.42 | 1.28 | 2.81 |
| 0.000064 | -8.09 | 1.25 | 2.95 |
| 0.000064 | -7.69 | 1.25 | 2.91 |
| 0.000064 | -7.80 | 1.25 | 2.92 |
| 0.000065 | -6.87 | 1.28 | 2.85 |
| | | | |
| 0.000066 | -6.39 | 1.29 | 2.81 |
| 0.000066 | -6.30 | 1.29 | 2.80 |
| 0.000066 | -6.35 | 1.29 | 2.80 |
| 0.000066 | -6.52 | 1.30 | 2.82 |
| 0.000066 | -6.37 | 1.29 | 2.81 |
| 0.000066 | -6.14 | 1.29 | 2.79 |
| 0.000065 | -5.46 | 1.26 | 2.75 |
| 0.000065 | -5.74 | 1.27 | 2.75 |

Ma et al. (2023b)

| -6.51 | 0.20 | (n =41, calculated by IsoplotR) |
|---|---|---|

$(^{143}Nd/^{144}Nd)_s = (^{143}Nd/^{144}Nd)_0 + (^{147}Sm/^{144}Nd)_s \times (e^{\lambda t} - 1)$
$\varepsilon_{Nd}(t) = [(^{143}Nd/^{144}Nd)_t/(^{143}Nd/^{144}Nd)_{CHUR(t)} -1] \times 10000$
$T_{DM2}=1/\lambda \times Ln\{1+[(^{143}Nd/^{144}Nd)_{DM} -(^{143}Nd/^{144}Nd)_S+((^{147}Sm/^{144}Nd)_S-(^{147}Sm/^{144}Nd)_{CC}) \times (e^{\lambda t}-1)]$
$\varepsilon_{Nd}(t) = [(^{143}Nd/^{144}Nd)_t/(^{143}Nd/^{144}Nd)_{CHUR(t)} -1] \times 10000/((^{147}Sm/^{144}Nd)_{DM} -(^{147}Sm/^{144}Nd)_{CC})\}$
$\lambda_{147Sm} = 0.654 \times 10^{-11}/a^{-1}$
$^{143}Nd/^{144}Nd)_{DM}=0.51315$
$^{147}Sm/^{144}Nd)_{DM}=0.2137$
$^{147}Sm/^{144}Nd)_{CC}=0.12$
Error of initial ratio is calculated from the measurement error of the isotope ratio, the estimated
concentration error and the age error. The decay constant is considered to be a fixed value.
$\sigma_{Nd(t)}$ is mean-square deviation of $(^{143}Nd/^{144}Nd)_t$
$\sigma_{Sm}$ is mean-square deviation of $(^{143}Sm/^{144}Nd)_s$
$\sigma_t$ is mean-square deviation of age
$$\sigma_{Nd(t)} = \sqrt{\sigma_{Nd}^2 + \sigma_{Sm}^2(e^{\lambda t} - 1)^2 + \sigma_t^2(\lambda e^{\lambda t}(\frac{147_{Sm}}{144_{Nd}}))^2}$$



**Table 4** Whole-rock Pb isotopic compositions of the Jiguanshan diorite

| Spon.no | U (ppm) | Th (ppm) | Pb (ppm) | $^{206}Pb/^{204}Pb$ | ±2SE | $^{207}Pb/^{204}Pb$ | ±2SE |
|---------|---------|----------|----------|---------------------|--------|---------------------|--------|
| ZY2201 | 0.70 | 4.28 | 16.38 | 15.867 | 0.0005 | 15.189 | 0.0005 |
| ZY2202 | 0.98 | 6.43 | 21.20 | 16.167 | 0.0008 | 15.243 | 0.0009 |
| ZY2203 | 0.88 | 6.71 | 18.03 | 15.882 | 0.0006 | 15.182 | 0.0006 |
| ZY2204 | 0.71 | 4.27 | 16.29 | 16.097 | 0.0010 | 15.225 | 0.0009 |
| ZY2205 | 0.75 | 3.87 | 18.90 | 15.832 | 0.0007 | 15.179 | 0.0006 |
| ZY2206 | 0.61 | 3.22 | 15.22 | 15.914 | 0.0010 | 15.170 | 0.0010 |
| ZY2207 | 0.68 | 3.55 | 14.22 | 16.036 | 0.0008 | 15.199 | 0.0007 |





| $^{208}Pb/^{204}Pb$ | ±2SE | $^{206}Pb/^{204}Pb$ | $^{207}Pb/^{204}Pb$ | $^{208}Pb/^{204}Pb$ | $^{238}U/^{204}Pb$ | $^{232}Th/^{204}Pb$ | $^{232}Th/^{238}U$ |
|---|---|---|---|---|---|---|---|
| | | initial | initial | initial | μ | ω | |
| 36.502 | 0.0014 | 15.063 | 15.103 | 35.027 | 2.6 | 16.0 | 6.3 |
| 37.126 | 0.0022 | 15.295 | 15.150 | 35.392 | 2.8 | 18.8 | 6.8 |
| 36.494 | 0.0013 | 14.965 | 15.084 | 34.398 | 2.9 | 22.8 | 7.8 |
| 37.324 | 0.0023 | 15.271 | 15.137 | 35.825 | 2.6 | 16.3 | 6.2 |
| 36.046 | 0.0016 | 15.095 | 15.100 | 34.901 | 2.3 | 12.4 | 5.3 |
| 36.124 | 0.0024 | 15.164 | 15.090 | 34.939 | 2.4 | 12.9 | 5.4 |
| 36.338 | 0.0016 | 15.136 | 15.103 | 34.931 | 2.9 | 15.3 | 5.4 |

Initial Pb isotopic ratios are calculated back to 1780 Ma.



**Supplementary material/Appendix:**
**Table S1** Zircon U–Pb isotopic data for the Jiguanshan diorite obtained by the LA-ICP-MS
technique
**Table S2** Zircon trace element data for the Jiguanshan diorite obtained by the LA-ICP-MS
technique