# Peer review of "Petrogenesis and tectonic setting of late Paleoproterozoic diorites in the"

_EGUsphere, 2025_

## Author Response (AR1)

**Dear Prof. Dr. Irene Bianchi**

Thank you very much for the opportunity to revise our manuscript. We appreciate the reviewers for their constructive comments and suggestions on the manuscript entitled "**Petrogenesis and tectonic setting of late Paleoproterozoic diorites in the Trans-North China Orogen**" by Wang Z et al. (No. egusphere-2025-2115).

We have carefully checked your and reviewers' comments and have revised the original manuscript following your and reviewers' suggestions. We hope that you will find our revised manuscript suitable for publication. All authors have seen the revised manuscript and agree with re-submission.

In the revised manuscript, we have re-organized the "Abstract", "Introduction" and "Discussions" sections. We have added more description of the basements, such as the Xiong'er and Taihua groups. Figures (such as Figures 1 and 2) are modified too. The major modifications in the revised manuscript are marked in Revisions Mode of Microsoft Word. Our responses to the comments and suggestions are given in the file "ResponseToReviewers".

Thank you and best regards.

Sincerely Yours,

Wang Z., He J., & co-authors

Dear Editors and Reviewers,

We would like to express our sincere gratitude for taking the time to review and evaluate our manuscript. The suggestions from the reviewers were extremely insightful and constructive. We have carefully considered each comment and have made the corresponding revisions to enhance the quality of the manuscript. Please find below our detailed explanations.

We sincerely hope that our research meets your expectations for publication. All authors have reviewed the revised manuscript and agree with its resubmission.

Yours sincerely,

Zhiyi Wang and Jun He on behalf of the other authors

**Response to reviewer**

**Reviewer 1:**

1. Please rearrange the abstract. Key data such as ages of the investigated diorite are not presented. The first sentence is also misleading, the Xiong'er volcanic rocks are not the focus of this study. The inferred conclusions are not supported by data. Please present more data patterns in the abstract.

We have revised the abstract according to your comments to better highlight the key work and contributions of this study.

2. Rearrange the Introduction

The scientific questions mentioned in the introduction are not clearly presented, and not closely linked with your discussion. Verifying the necessity of your research is very important. It would be better if you can slightly revise the introduction, or perhaps you can revise the discussion.

We have modified it accordingly. The introduction has been revised to better highlight the key issues addressed in this paper.

3. The research background should be more detailed. Give us more information on the assembly of the north China craton, related metamorphic and igneous records, as well

as related structures. Tell us more about the Xiong'er group, the Taihua group, their relations.

We have substantially enhanced the research background section with expanded discussions on: (1) the assembly processes of the North China Craton; (2) the models for the distribution of the tectonic units of the North China Craton; and (3) detailed information of the Xiong'er and Taihua Groups. These revisions will provide clearer context for our work.

4. I believe that models for the distribution of the tectonic units of the north China craton are very much varied. There are some other models that are not mentioned by the article, but should be mentioned.

We have modified it accordingly.

5. line 38 what is the ancient basement rocks? Be specific

We have modified it accordingly.

6. line 40-41, not logically correct.

We have modified it accordingly.

7. I don't see strong evidence of rifting setting. Please expand your discussion on the tectonic settings.

If the diorites and the Xiong'er groups were coeval, but the Xiong'er volcanics rocks were formed in an arc setting. How can you explain the within plate setting of these diorites.

In the revised manuscript, we have incorporated additional geological background information. At the beginning of the "Tectonic Setting" section, we explicitly evaluated whether the diorite samples exhibit typical geochemical features of intermediate rocks from convergent plate margins. Our results do not support a subduction-related origin. Building on these findings, we further discuss the petrogenesis of the diorites in relation to the Xiong'er volcanic rocks, and place them within the regional tectonic evolutionary framework.

8. Figure 1, revise.

We have updated Figures 1 to ensure the figure reflect the most current understanding

of the regional geology.

9. Figure 2, any good field photo?

Thank you for the suggestion. We've enhanced the field photographs.

**Reviewer 2:**

1. Lines 30-33, I agree with that the studied diorite formed in a rift setting. However, it is hard to say they mark a crustal-origin rock shift from orogenic-related magmatism to intraplate magmatism. How to define or constrain a tectonic transition? On the other hand, I suggest the authors distinguish between rift (intraplate) and post-collisional settings, as the latter is continuation of a collisional setting, but the former is not.

We have replaced the phrase "mark a shift" with the more cautious formulation "could be indicative of a potential transition". On one hand, we emphasized the transitional geochemical characteristics of the diorites. On the other hand, we supplemented the regional tectonic background and evolutionary history in the text, highlighting the significance of the diorites. These revisions have been made to clarity of our argument.

2. Introduction part, lines 42-47, since the previous research has been very clear that "...the craton experienced multiple rift phases, with the Xiong'er rift being the first rift formed after the assembly, resulting in the formation of the c. 1780 Ma Xiong'er volcanic rocks and contemporaneous mafic dyke swarms", why do the origin and tectonic setting of the Xiong'er volcanic rocks remain controversial? What is the crux or cause of these dispute? Please add related content.

We have modified it accordingly. We have expanded the geological background and cause of these disputes in the revised manuscript.

3. Line 83, please change "fault"to "Fault".

We have modified it accordingly.

4. Fig.1, the legend does not correspond to the pattern in the figure. Please recheck.

Lines 185-191, please add MSWD values for the calculated ages.

We have modified it accordingly.

5. Besides, the zircon spots with ages of ca. 1.6 Ga and 1.93 Ga may not belong to the same population as the zircons with ages of ca. 1.77-1.74 Ga. So they should be discussed separately instead of being taken into account.

We used zircon trace element analysis to constrain the tectonic setting. Outliers identified by Isoplot were excluded during data processing. All analytical spots with La contents >1 ppm were removed to ensure data reliability. Some zircons show evidence of Pb loss in Fig. 4. Nevertheless, the average $^{207}Pb/^{206}Pb$ ages still exhibit a relatively concentrated distribution in Fig.4. Few outliers cannot undermine the overall coherence. Furthermore, all diorite samples in this study were processed using consistent data processing, and the large number of valid data points significantly reduces the potential impact of individual anomalies on the general interpretation.

6. Line 200, add "wt. %"after 5.57, i.e., from 5.57 wt. % to 6.03 wt. %." Please check the whole text for the similar problems.

We have modified it accordingly.

7. Line 240, the SiO2 contents are different from those in line 199, why?

In the Results section, the contents described refer specifically to the samples from the Jiguanshan diorite. In the Discussion, we compare them with the compositional ranges of regional diorites of similar age.

8. Discussions on the sources of the diorite, such as lines 315-320, I recommend to use the isotopic data first, which is more effective to eliminate relevant source regions.

We provide a comprehensive analysis of the source of the diorites by integrating multiple lines of evidence, including elemental geochemistry and isotopic compositions. By comparing isotopic characteristics with those of potential source reservoirs, such as the basement rocks of the Taihua Group, the contemporaneous Xiong'er volcanic rocks, regional mafic dike swarms, and crustal rocks' characteristics, we conclude that these diorites were derived from partial melting of middle to lower crustal rocks.

9. Line 368, please delete the superfluous ".".

We have modified it accordingly.

10. Lines 423-426, in fact, most of the samples fell into the rift/ anorogenic environment fields, consistent with the previous discussion. But why do the author say "These may indicate that the post-collisional extension during this period may ultimately lead to rift evolution continuously and progressively." How and when lead to?

We have enhanced the discussion by incorporating additional regional geological context and a more detailed analysis of the tectonic evolution. Following the ~1.85 Ga collisional event, the North China Craton entered a prolonged post-collisional extensional stage. Initial crustal thickening and remelting generated extensive crust-derived granites. Subsequent slab breakoff and gravitational collapse induced mid- to upper-crustal extension and felsic magmatism. By ~1.78 Ga, continued lithospheric thinning led to asthenospheric upwelling and melting of subduction-modified lithospheric mantle. The magmatism then evolved into A-type granites and alkaline rocks, marking a transition to an anorogenic intracontinental extensional setting. The 1.78 Ga crust-derived diorites exhibit transitional features, retaining orogenic signatures while shifting toward an intraplate affinity, reflecting prolonged extension after craton amalgamation.

11. About the tectonic setting, similar to comment 1, it is still necessary to distinguish whether the ca. 1.78 Ga diorite was formed in a rift or a post-collision environment? Meanwhile, please comprehensively define the tectonic background in combination with other coeval rock combinations (e.g., granitoids) and research cases of the same period.

In the revised manuscript, we have enhanced the regional geological background, summarized tectonic models and debates related to other coeval rock. Furthermore, we have added a concluding section discussing the tectonic evolution and significance of the diorites. These revisions improve the clarity of our arguments.

---

## Author Response (AR3)

**Dear Prof. Dr. Chu and Editors,**

We sincerely appreciate your professional suggestions and diligent efforts, which have been essential in improving the quality and accuracy of our work. We have addressed all of your suggestions in the manuscript. We have removed the content on page 58 of the previous PDF file as requested and the newly uploaded version is clean and without any colorful text but marked charges.

We sincerely hope that our research meets your expectations for publication. All authors have reviewed the revised manuscript and agree with its resubmission.

Yours sincerely,
Zhiyi Wang and Jun He on behalf of the other authors